# The Impact of Non-Motor Symptoms on Quality of Life in Cervical Dystonia

**DOI:** 10.3390/jcm12144663

**Published:** 2023-07-13

**Authors:** Raffaela Maione, Caterina Formica, Angelo Quartarone, Viviana Lo Buono

**Affiliations:** IRCCS Centro Neurolesi “Bonino-Pulejo”, 98124 Messina, Italy; raffaela.maione15@gmail.com (R.M.); angelo.quartarone@irccsme.it (A.Q.); viviana.lobuono@irccsme.it (V.L.B.)

**Keywords:** cervical dystonia, non-motor symptoms, quality of life

## Abstract

Cervical dystonia (CD) is characterized by cranial muscle overactivity leading to abnormal intermittent or continuous posturing of the head. Nowadays, the treatment of patients suffering from this condition focuses principally on the motor component of the disorder, certainly the invaliding part; however, it leaves out the non-motor one that has a similarly invalidated effect on the quality of the subject’s life. This review was conducted on studies investigating the impact of non-motor symptoms on levels of quality of life. We searched on the PubMed, EMBASE and Web of Science databases and screening references of included studies and review articles for additional citations. From an initial 150 publications, we included only five studies that met the search criteria. The results showed that anxiety, depression, pain and sleep quality have a great influence on patients’ health and on the outcome of the disease. Future studies should focus more on investigating the non-motor components of CD as an integral part of the clinical management of dystonic patients in order to improve their well-being.

## 1. Introduction

Dystonia is a neurological condition characterized by sustained or intermittent muscle contractions causing abnormal, often repetitive, movements, postures or both [1]. Dystonic movements are typically prolonged and stereotypical and assume a torsional appearance (a feature that distinguishes dystonic movements from other dyskinesias) and are usually associated with the simultaneous contraction (cocontraction) of agonist and antagonist muscles; they are exacerbated by fatigue and stress, may be present at rest and worsen with voluntary movement. The current classification of dystonia is based on: Clinical features: Age at onset, affected areas of the body, time course (pattern), coexistence of other movement disorders, coexistence of other neurological or systemic manifestations.Etiology: Pathology of the nervous system, inheritance and other acquisitions of the disease [1,2].

Depending on the regions of the body affected, dystonia can be subclassified as follows:

Focal dystonia: Abnormal movements involving a single region of the body (blepharospasm, writer’s cramp).

Segmental dystonia: Abnormal movements affecting two or more adjacent parts of the body (e.g., craniocervical dystonia, Meige syndrome).

Generalized: Abnormal movements affecting different regions of the body [3,4].

Cervical dystonia (CD) is the most prevalent form of dystonia [5]. It is a condition characterized by cranial muscle overactivity leading to abnormal intermittent or continuous posturing of the head. The activation of various cervical muscles gives rise to multiple types of neck posturing (retrocollis, anterocollis, torticollis, laterocollis) [6]. The anatomical basis for CD is still unclear. Some studies have suggested that CD is caused by a disruption of the basal ganglia due to the identification of patients with discrete focal lesions (often in putamen) who had dystonia. In addition, in patients with CD, cortical alterations have also been found in other structures such as the thalamus, the brain stem and the cerebellum [7]. However, subsequent studies have shown that dystonia is a network disorder involving a basal ganglia–cerebello–thalamo–cortical circuit [8].

Although still considered a paradigmatic movement disorder, CD is associated with a broad spectrum of non-motor symptoms (NMSs) including abnormalities in sensory and perceptive functioning, cognitive and mood alterations and sleep disorders [9]. As with motor symptoms, NMSs may substantially and profoundly affect quality of life (QoL) in CD, consequently, it is essential to recognize its symptoms and provide appropriate management to improve patient QoL [10]. QoL is a subjective issue related to the person’s physical health, psychological state, level of independence, social relationship, person beliefs and relationship with the environment [11,12]. Health related quality of life (HRQoL) evaluates how the individual’s well-being might be affected over time by disease, disability or disorder [13,14]. HRQoL is understood as the perception and evaluation by the patients themselves of the impact caused in their lives by the disease and its consequences; it includes physical, mental and social aspects related to the state of health and care, in addition to the global perceptions about health and other personal constructs [15].

NMSs are very frequent reported in CD patients [16]. Evidence suggests that NMSs are the most important predictors of decreased HRQoL in patients with CD since they cause an impairment of the activities of daily living, and they are described independently by motor severity [17,18]. In contrast, other authors have reported NMSs in CD in association with major motor disability [19]. It has also been observed that NMSs, in particular neuropsychiatric symptoms and pain, are more important predictors of QoL than the severity of dystonia itself [20,21,22].

This systematic review focused on the studies that investigated the impact of NMSs on QoL in patients with CD.

## 2. Methods

### Search Strategy

This systematic review was conducted on studies investigating the impact of NMSs on the quality of life in CD. Studies were identified by searching on MEDLINE, EMBASE and Web of Science databases from 2013 to 2023 with the following search keywords: (“torticollis” [MeSH Terms] OR “torticollis” [All Fields] OR (“cervical” [All Fields] AND “dystonia” [All Fields]) OR “cervical dystonia” [All Fields]) AND (“quality of life” [MeSH Terms] OR (“quality” [All Fields] AND “life” [All Fields]) OR “quality of life” [All Fields]).

The search terms were identified as title and abstract. Only English texts and studies on human subjects were selected. After the duplicates were removed, all articles were evaluated based on the title, abstract and text. All studies that examined the relationship between NMSs and QoL in patients with CD were included, after they fulfilled the following criteria:

Published peer-review research.

The sample population included patients with CD.

Studies specifically assessed the relationship between NMSs and QoL.

Studied using standardized test.

We excluded case studies.

## 3. Results

Of the 150 identified, only five studies met the inclusion criteria (Figure 1). All studies conducted research on 525 CD patients and examined the association between QoL and NMSs (Table 1). Four different measures of QoL (Table 2) were identified: Craniocervical dystonia questionnaire (CDQ-24) [23], 36-item short form health survey (SF-36) [24], Cervical Dystonia Impact Profile (CDIP-58) [25] and EuroQol Utility Values (EQ-5D-5L) [26]. The quality assessment of studies was performed with the National Institute for Health and Care Excellence (NICE, 2010) guidelines.

### Measure of Quality of Life

Many data in the literature have focused on NMSs in dystonia; however, only a few studies have investigated the impact of the disease on the quality of life of the patients. 

Werle et al. [11] conducted research on 70 patients with CD to investigate in detail which way the physical, emotional and social aspects are affected and compromising the QoL using the Craniocervical dystonia questionnaire (CDQ-24) [23]. The results described an interference of CD in performing the activities of daily living; in particular, patients showed signs of isolation, impaired social interaction and emotional limitation that worsened their QoL.

Stamelou et al. [31] focused on the prevalence and severity NMSs, such as fatigue, excessive daytime sleepiness (EDS) and sleep quality, depression and anxiety and their influence on the QoL in patients with CD compared to a control group.

HRQoL in patients with CD not only is influenced by psychiatric comorbidity, but fatigue and excessive daytime sleepiness and pain also appeared to be significant contributors to a decreased perceived well-being.

A similar study was conducted by Tomic et al. [28] on the presence of psychiatric symptoms and anxiety in patients with CD and their impact on QoL. They recruited 19 patients and have administered the Toronto Western Spasmodic Torticollis Rating Scale (TWSTRS) [1] to evaluate the symptoms of dystonia, Beck Depression Inventory II (BDI-II) and Beck Anxiety Inventory (BAI) to assess depression and anxiety, the Craniocervical Dystonia Questionnaire (CDQ-24) and the short version of Health Survey (SF-36) to analyze QoL. The results showed that depression and anxiety, but also stigma, emotional state, pain, daily activity and social life, worsened the QoL of the patients.

Han et al. [29] focused on the association between NMSs and QoL in 102 patients with CD. Participants have completed the following questionnaire: Toronto Western Spasmodic Torticollis Rating Scale (TWSTRS) [1], 7-point Clinical Impression Scale (CGI-S) [32]; Beck Depression Inventory II and Beck Anxiety Inventory (BAI); Starkstein’s Apathy Scale (SAS); Multidimensional Fatigue Inventory (MFI); Pittsburgh Sleep Quality Index (PSQI); and the Short Form Health Survey (SF-36). This research delineated an important prevalence of NMSs in patients with CD; in particular EDS, poor sleep, depression and fatigue were determinants in HRQoL. In addition, Monaghan et al. [30] assessed the inter-relationships between NMSs and HRQoL on 46 participants by four questionnaires: Cervical Dystonia Impact Profile (CDIP-58) [25], EuroQol Utility Values (EQ-5D-5L) [26], Beck Anxiety Inventory (BAI) and Beck Depression Inventory II (BDI-II). Lower levels of HRQoL were associated with higher levels of psychological distress and pain, stressing the importance of NMSs in the treatment of CD. The majority of participants with CD had clinically significant levels of anxiety and depression; these psychological distresses were not associated with the severity of CD motor symptoms, while pain had a greater impact on them.

## 4. Discussion

CD is a chronic disease associated with a reduction in HRQoL [33]. Patients with CD report, indeed, several problems with daily activities, mobility and self-care that negatively impact their QoL [34]. In recent years, the evaluation of QoL has increasingly been recognized as important in the assessment and management of this disorder [20,35]. Although the most important factors influencing QoL in CD patients seems to be impaired physical functioning due to the presence of motor symptoms [36,37], there is growing interest on the influence of NMSs on subjective well-being. CD, indeed, is presenting with several NMSs that affect the global clinical status of patients. The results of this review show how CD patients face a life of chronic disability and associate their reduce QoL with factors involving both their physical and emotional health [34,36]. Among NMSs, psychological disorders are frequently reported in CD and its influence on QoL and the self-perceived response to various treatments [38,39,40]. Depression and anxiety, in particular, generally appear to be associated with low HRQoL regardless of the severity of motor symptoms [29]. In addition, the most frequently reported emotional problems are related to the stigma linked to the disease and to a feeling of insecurity around new people, problems with friends and family, fears regarding the disease and feelings of sadness [34,36]. A significant percentage of patients report difficulties in performing recreational activities, meeting job demands, difficulty in attending public places, discomfort in public, having concerns with people’s reaction to illness and isolation [20]. The stigma is associated with the feeling of body deformation and probably favors the onset of mood alterations.

HRQoL in CD patients is not only affected by stigma or psychiatric comorbidity [11], but also by fatigue and excessive daytime sleepiness. Smit and al. [27] highlight the need for a systematic screening of fatigue and excessive daytime sleepiness in daily practice and the need to treat possible factors that contribute to a worsening of patients’ QoL such as depression and pain. Indeed, pain as a stress factor, acting in conjunction with other factors, contributes to increasing the likelihood of depression [39] that is the main predictor of worsened QoL in CD patients [21]. The prevalence of impaired sleep quality in patients with cervical dystonia is between 40% and 70% [41]. Sleep disorders are very common in patients with cervical dystonia, and they often occur in comorbidity with anxiety and depression, significantly correlating with poor quality of life [42]. Some research has shown that pain can also contribute to the development of sleep disorders in CD, particularly in the transition to the first phase of sleep [43]. Sleep impairment is particularly disabling for CD patients. Several studies using “Cervical Dystonia Impact Scale” to measure the impact of cervical dystonia on health based on patient perceptions [25], showing how a large percentage of subjects report a negative impact of sleep on QoL [44].

## 5. Conclusions

This review focused on the lack of studies that explore the relationship between the impact of NMSs on QoL in CD. A small number of works were included in this review since only five studies met the inclusion criteria. Furthermore, a meta-analysis was unable to be performed because quantitative information was not reported in the included studies. None of the reviewed studies employed a longitudinal design, and this is a limitation related to cross-sectional design in this research area.

The overall quality of evidence was low and we observed a significant weakness in the definition of QoL and a methodological variability in the qualitative and quantitative measures of HRQoL. The studies examined suggested that depression and anxiety are directly related with the severity of motor symptoms. Consequently, the reported emotional problems could be related to the stigma linked to the disease and to feelings of insecurity and isolation [20], and feelings of body deformation that probably favor the onset of depressive symptoms. A limitation is that potentially confounding factors such as mood comorbidity, anxiety and depression, pain, fatigue were not quantitatively measured. In addition, the drug treatments were not adequately described.

Despite these limitations, this descriptive review underlines important implications in the clinical management of CD.

Often, the focus is on the motor component of the disorder, leaving out the set of non-motor characteristics that are equally part of the pathophysiological picture of dystonia. The NMSs need a lot of attention from a clinical point of view, as they strongly influence the QoL and play a fundamental role in the perception of the patient, both of himself and of his life in general [10]. In addition, NMSs are an interesting starting point for research thanks to studies already carried out that classify a part of the symptoms analyzed as a primary component of dystonia [45].

Multicentric future studies should focus more on investigating the non-motor components related to the CD, such as anxiety, depression and stress, deepening the study of neurocognitive models and improving accessibility to neuropsychological assessments. From the studies reviewed, we realized that no standardized neuropsychological battery has been used for the assessment of non-motor disorders in dystonia. Therefore, it would be useful to identify tests that could give a more complete point of view about the presence of cognitive deficits and psychological disorders so that we would have a quantitatively measurable figure. For this reason, active screening for the non-motor spectrum and subsequent treatment should be an integral part of the clinical management of dystonic patients in order to improve their well-being. A comprehensive view of a patient’s well-being should become a guiding concept for the treating clinician and a therapeutic trial outcome measure for patients.

## Figures and Tables

**Figure 1 jcm-12-04663-f001:**
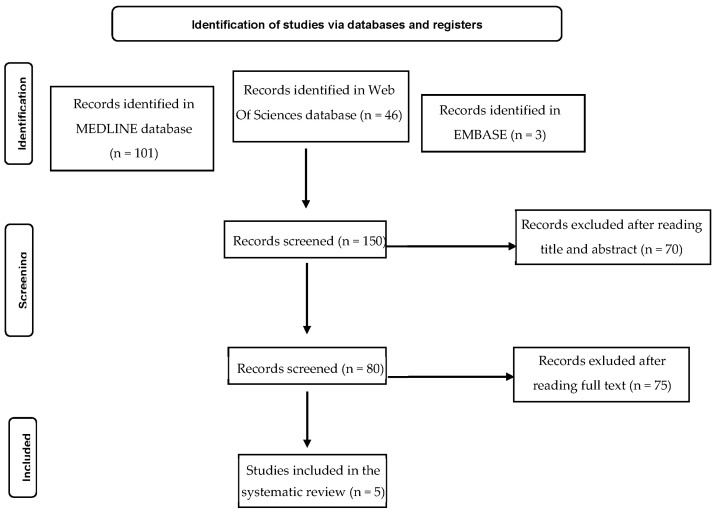
PRISMA flow diagram.

**Table 1 jcm-12-04663-t001:** Studies (5) assessing the influence of NMSs on QoL.

Study	Aim	Sample Size	Socio-Demographic Characteristics	Assessment	Outcomes	Results
Werle et al., 2014 [11]	Verify the existence of a correlation between the level of motor impairment, pain and QoL	70 CD patients	50 years old, range 21–79 years, being 44 (63%) women and 26 (37%) men Median age of onset: 39 years old, range 0–76 years	CDQ-24	The physical, social and emotional aspects are the most affected in the QoL	Median = 52 (39–66 (first and third quartiles) points ranging between 13 and 90 points.
Smit et al., 2016 [27]	Examined the prevalence and severity of fatigue, excessive daytime sleepiness and sleep quality	44 CD patients and 43 controls	51 years old, range 20–80 years Mean age of controls: 54 years old, range 25–83 years	FSS; ESS; PSQI; BDI-II; BAI; RAND-36	Pain and motor severity, fatigue, worse QoL	BDI = 10.6 ± 7.3 4.5 ± 5.0 <0.01 BAI = 9.3 ± 6.8 4.0 ± 4.2 <0.01 FSS = 4.4 ± 1.7/4.0 2.7 ± 1.4/3.1 <0.01/0.01 ESS = 8.8 ± 6.9/7.3 5.8 ± 4.9/7.4 0.04/0.95 PSQI = 7.4 ± 3.9/6.5 5.1 ± 4.4/6.1 <0.01/0.73
Tomic et al., 2016 [28]	Analyze the presence of depression and anxiety in CD	19 CD patients	11 females and 8 males with mean ages of 59.37 ± 12.96 (age range 30–79) years Mean disease duration: 9 ± 6.46 (range 1–24) years	TWSTRS; BDI-II; BAI; CDQ-24; SF-36	Mild depression and moderate anxiety consequences on QoL	TWSTRS = 23.89 ± 9.51 CDQ-24 subscales = Disability yielded highest correlation with pain (r = 0.765) and daily activity (r = 0.755) SF-36 = Disability correlated mostly with physical function (q = 0.684) and emotional disability (q = 0.654), while pain correlated mostly with body pain (q = 0.744) and physical function (q = 0.636)
Han et al., 2020 [29]	Assess the prevalence of depression, anxiety, fatigue, apathy, pain, sleep problems and EDS in CD patients	102 CD patients	76 females and 26 males with mean ages of 55.6 ± 13.9 years Mean age of dystonia onset: 44.9 ± 15.1 years Mean duration of the disease: 9.9 ± 8.9 years	TWSTRS; CGI-S; BDI-II; BAI; SAS; MFI); PSQI; SF-36.	Poor sleep, depression and fatigue seem to be determinants in HRQoL in CD	CGI-S = mean (SD) 4.2 (1.2) Median (range) = 4 (2–7) TWSTRS = Mean (SD) 16.4 (5.3) Medina (range) 16 (6–30) BDI-II = mean (SD) 14.4 (10.6) median (range)12 (0–42) BAI = mean (SD) 15.1 (10.3) median (range)14 (0–49) SAS = mean (SD) 12.1 (6.8) median (range)11 (1–30) MFI = mean (SD)13.4 (4.9) median (range)14 (4–20) PSQI = mean (SD) 6.4 (3.7) median (range) 5 (0–18) SF-36 Physical health = mean (SD) 39.5 (10.2) SF-36 Mental health= mean (SD) 41.1 (12.1)
Monaghan et al., 2020 [30]	Assess cognition in CD patients and the interrelationships between NMSs and HRQoL	46 CD participants	31 females and 15 males with mean ages of 68 ± 10.7, range 33–80	CDIP-58; EQ-5D-5L; BAI; BDI-II.	Pain and psychological distress were associated with low HRQoL	HADS anxiety = M ± SD 7.9 ± 4.83 HADS depression = M ± SD 4.61 ± 3.67 BAI = M ± SD 9.48 ± 9.5 BDI = M ± SD 10.32 ± 10.91 CDIP-58 Total Score = M ± SD 30.41 ± 20.83

Legends: CDQ-24 = Craniocervical dystonia questionnaire; FSS = Fatigue Severity Scale; ESS = Epworth Sleepiness Scale; PSQI = Pittsburgh Sleep Quality Index; BDI-II = Beck Depression Inventory; BAI = Beck Anxiety Inventory; RAND-36 = RAND-36 item Health Survey; TWSTRS = Toronto Western Spasmodic Torticollis Rating Scale; SF-36 = short version of Health Survey; CGI-S = 7-point Clinical Impression Scale; SAS = Starkstein’s Apathy Scale; MFI = Multidimensional Fatigue Inventory; CDIP-58 = Cervical Dystonia Impact Profile; EQ-5D-5L = EuroQol Utility Values; HADS-A = Hospital Anxiety and Depression Scales.

**Table 2 jcm-12-04663-t002:** Measures used for the assessment of QoL in CD patients.

Questionnaire	Domains	Items	Scales	Focus
Craniocervical dystonia questionnaire (CDQ-24)	5 domains: stigma, emotional wellbeing, pain, activities of daily living and family/social life	24 items	5-point scale (from 0 to 4). The total score ranges from 0 (best QoL) to 100 (worst QoL)	To addresses the perceptions and concerns of the patients with craniocervical dystonia to assess the impact of the disease
36-item short form health survey (SF-36)	8 domains: limitations in physical activities because of health problems, limitations in social activities because of physical or emotional problems, limitations in usual role activities because of physical health problems, bodily pain, general mental health, limitations in usual role activities because of emotional problems, vitality, general health perceptions	11 items	5-point scale (from 1 to 5) or 2-point scale (from 1 to 2) or 3-point scale (from 1 to 3) or 6-point scale (from 1 to 6) or The items are summed up to give 0–100 scores. Lower results represent worse QoL	To assess health at the individual level in clinical practice and research and at the population level for health policy evaluations, and general population survey
Cervical Dystonia Impact Profile (CDIP-58)	8 domains: head and neck symptoms, pain and discomfort symptoms, upper limb activities, walking, sleep, annoyance, mood, psychosocial functioning	58 items	5-point scale (from 1 to 5). The eight summary scale scores are generated by summing items and then transformed to a 0–100 scale. High scores indicate worse health	To assess patients’ perceptions and measure the health impact of cervical dystonia on patients’ lives
EuroQol Utility Values (EQ-5D-5L)	5 domains: mobility, self-care, usual activities, pain/discomfort and anxiety/depression.	26 items	5-point levels: no problems, slight problems, moderate problems, severe problems and extreme problems. Visual Analogical Scale (VAS) 0–100	To assess health related quality of life

## Data Availability

Data available on request from the authors.

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
