# Peer review of "The Impact of Non-Motor Symptoms on Quality of Life in Cervical Dystonia"

_jcm, 2023, doi:10.3390/jcm12144663_

Round 1
Reviewer 1 Report
The paper of Maione et al "The impact of non-motor symptoms on quality of life in cervical dystonia" concerns an interesting topic and well set-out. Overall the paper is good and delivers an important message.
My main suggestion for improvement is for the authors look critically at the source studies to see whether there was any attempt to evaluate NMS severity and impact on QoL independent of motor severity of cervical dystonia.
Also suggest the authors comment on which NMS contribute most significantly to reduced QoL: is it pain, distorted body image/self consciousness, anxiety, depression, sleep disturbance? This may be impossible to tell from the source studies but some comment would be useful.
There some errors of english but these should be easily corrected in final editing.
eg Abstract line "invaliding" is incorrect do the authors mean "invalidating"
Line 1 Introduction the word "condition" is missing after "neurological"
Line 5 Conclusion "of the" is missing after "None" and "not" should be deleted after "studies" as it is a double negative
Author Response
REVIEWER 1
Open Review
( ) I would not like to sign my review report
(x) I would like to sign my review report
Quality of English Language
( ) I am not qualified to assess the quality of English in this paper
( ) English very difficult to understand/incomprehensible
( ) Extensive editing of English language required
( ) Moderate editing of English language required
(x) Minor editing of English language required
( ) English language fine. No issues detected
|
Yes |
Can be improved |
Must be improved |
Not applicable |
|
|
Does the introduction provide sufficient background and include all relevant references? |
(x) |
( ) |
( ) |
( ) |
|
Are all the cited references relevant to the research? |
(x) |
( ) |
( ) |
( ) |
|
Is the research design appropriate? |
(x) |
( ) |
( ) |
( ) |
|
Are the methods adequately described? |
(x) |
( ) |
( ) |
( ) |
|
Are the results clearly presented? |
(x) |
( ) |
( ) |
( ) |
|
Are the conclusions supported by the results? |
(x) |
( ) |
( ) |
( ) |
Comments and Suggestions for Authors
The paper of Maione et al "The impact of non-motor symptoms on quality of life in cervical dystonia" concerns an interesting topic and well set-out. Overall the paper is good and delivers an important message.
My main suggestion for improvement is for the authors look critically at the source studies to see whether there was any attempt to evaluate NMS severity and impact on QoL independent of motor severity of cervical dystonia.
Also suggest the authors comment on which NMS contribute most significantly to reduced QoL: is it pain, distorted body image/self consciousness, anxiety, depression, sleep disturbance? This may be impossible to tell from the source studies but some comment would be useful.
Comments on the Quality of English Language
There some errors of english but these should be easily corrected in final editing.
eg Abstract line "invaliding" is incorrect do the authors mean "invalidating"
Line 1 Introduction the word "condition" is missing after "neurological"
Line 5 Conclusion "of the" is missing after "None" and "not" should be deleted after "studies" as it is a double negative
Response author: thank you for your suggestions. I correct all this grammatical and English language errors. In conclusion section I added a sentence to explain better the concept that you suggest about comment on which NMS contribute most significantly to reduced QoL.
Reviewer 2 Report
Despite data found regarding non-motor symptoms in patients with cervical dystonia are not enough for a meta-analysis, the authors should describe quantitative data of the papers included in the review instead of a mere repetition of the conclusions of the authors of these papers. For example, they could make an estimative analysis on the scores of PSQI, SF-36 and other scales that were used in more than one of these papers by combining the data of al least 2-3 works. The did not describe quantitative data of patients compared with controls in the unique publication that included heatlthy controls.
The authors could propose how they consider an adequate design for future studies. (multicenter?, prospective, types of scales uses for non-motor symptoms, etc).
Minor comment: in the introduction Dystonia is a neurological....... syndrome?, disorder?..
Author Response
REVIEWER 2
Open Review
( ) I would not like to sign my review report
(x) I would like to sign my review report
Quality of English Language
(x) I am not qualified to assess the quality of English in this paper
( ) English very difficult to understand/incomprehensible
( ) Extensive editing of English language required
( ) Moderate editing of English language required
( ) Minor editing of English language required
( ) English language fine. No issues detected
|
Yes |
Can be improved |
Must be improved |
Not applicable |
|
|
Does the introduction provide sufficient background and include all relevant references? |
(x) |
( ) |
( ) |
( ) |
|
Are all the cited references relevant to the research? |
(x) |
( ) |
( ) |
( ) |
|
Is the research design appropriate? |
( ) |
(x) |
( ) |
( ) |
|
Are the methods adequately described? |
( ) |
(x) |
( ) |
( ) |
|
Are the results clearly presented? |
( ) |
(x) |
( ) |
( ) |
|
Are the conclusions supported by the results? |
( ) |
(x) |
( ) |
( ) |
Comments and Suggestions for Authors
Despite data found regarding non-motor symptoms in patients with cervical dystonia are not enough for a meta-analysis, the authors should describe quantitative data of the papers included in the review instead of a mere repetition of the conclusions of the authors of these papers. For example, they could make an estimative analysis on the scores of PSQI, SF-36 and other scales that were used in more than one of these papers by combining the data of al least 2-3 works. The did not describe quantitative data of patients compared with controls in the unique publication that included heatlthy controls.
The authors could propose how they consider an adequate design for future studies. (multicenter?, prospective, types of scales uses for non-motor symptoms, etc).
Minor comment: in the introduction Dystonia is a neurological....... syndrome?, disorder?..
Response author: Dear reviewer thank you for your suggestions. This review was considered as a systematic review because data that we collected are not consistent for a meta-analysis. This is a limitation of our study that we described in conclusion section. I reported the sentence “Furthermore, a meta-analysis was unable to be performed because quantitative information was not reported in the included studies. None of the reviewed studies not employed a longitudinal design, and this is a limitation related to cross-sectional design in this research area”.
In conclusion section I added a sentence to explain better design fir future studies.
I correct the mistake and added the word “condition” after neurological in introduction section.
Round 2
Reviewer 2 Report
The authors have made some corrections, but it should be useful for the readers to add numerical data on the results of the studies included in the review (scores of PSQI, SF36, etc), that could be added in a column of the table, as it was commented in the first review.
Author Response
REVIEWER 2
Open Review
Open Review
( ) I would not like to sign my review report
(x) I would like to sign my review report
Quality of English Language
(x) I am not qualified to assess the quality of English in this paper
( ) English very difficult to understand/incomprehensible
( ) Extensive editing of English language required
( ) Moderate editing of English language required
( ) Minor editing of English language required
( ) English language fine. No issues detected
|
Yes |
Can be improved |
Must be improved |
Not applicable |
|
|
Does the introduction provide sufficient background and include all relevant references? |
(x) |
( ) |
( ) |
( ) |
|
Are all the cited references relevant to the research? |
(x) |
( ) |
( ) |
( ) |
|
Is the research design appropriate? |
( ) |
(x) |
( ) |
( ) |
|
Are the methods adequately described? |
(x) |
( ) |
( ) |
( ) |
|
Are the results clearly presented? |
( ) |
(x) |
( ) |
( ) |
|
Are the conclusions supported by the results? |
(x) |
( ) |
( ) |
( ) |
Comments and Suggestions for Authors
The authors have made some corrections, but it should be useful for the readers to add numerical data on the results of the studies included in the review (scores of PSQI, SF36, etc), that could be added in a column of the table, as it was commented in the first review.
Response Author: dear reviewer, I added a column in table 1 titled “results” where I inserted results for all article that we considered in our review, as you suggested. Thank you